# Colles’ Fracture: An Epidemiological Nationwide Study in Italy from 2001 to 2016

**DOI:** 10.3390/ijerph20053956

**Published:** 2023-02-23

**Authors:** Umile Giuseppe Longo, Sergio De Salvatore, Alessandro Mazzola, Giuseppe Salvatore, Barbara Juliette Mera, Ilaria Piergentili, Vincenzo Denaro

**Affiliations:** 1Research Unit of Orthopaedic and Trauma Surgery, Fondazione Policlinico Universitario Campus Bio-Medico, 200-00128 Roma, Italy; 2Research Unit of Orthopaedic and Trauma Surgery, Department of Medicine and Surgery, Università Campus Bio-Medico di Roma, 21-00128 Roma, Italy; 3School of Medicine, University of Miami Miller School of Medicine, Miami, FL 33136, USA

**Keywords:** Colles, fracture, burden, surgery, nationwide, registry, Italy, epidemiology

## Abstract

The present study aimed to evaluate the yearly number of Colles’ fractures in Italy from 2001 to 2016, based on official information found in hospitalization records. A secondary aim was to estimate the average length of hospitalization for patients with a Colles’ fracture. A tertiary aim was to investigate the distribution of the procedures generally performed for Colles’ fractures’ treatment in Italy. An analysis of the National Hospital Discharge records (SDO) maintained at the Italian Ministry of Health, concerning the 15 years of our study (from 2001 to 2016) was performed. Data are anonymous and include the patient’s age, sex, domicile, length of hospital stays (days), primary diagnoses and primary procedures. From 2001 to 2016, 120,932 procedures for Colles’ fracture were performed in Italy, which represented an incidence of 14.8 procedures for every 100,000 adult Italian inhabitants. The main number of surgeries was found in the 65–69- and 70–74-year age groups. In the present study, we review the epidemiology of Colles’ fractures in the Italian population, the burden of the disease on the national health care system (in terms of length of hospitalization) and the distribution of the main surgical procedures performed for the treatment of the disease.

## 1. Introduction

Distal radius fractures (DRFs) are the most common injury in the adolescent and adult population in the Western world [1,2,3]. DRFs in younger patients are most commonly associated with high-energy trauma [4], while low-energy trauma is the most likely causative mechanism of injury in older patients [5]. Moreover, risk factors in the young population include sporting activities and motor vehicle accidents. Instead, obesity and osteoporosis are common in injured elderly patients [6,7,8]. The management of DRFs ranges from applying a bandage to open surgery [3]. However, the optimal treatment is still debated and depends on both the fracture and patient features [9,10].

DRFs are usually associated with poor quality of life, high healthcare costs and reduced working activities [11]. Most patients fully recover within 3–6 months, but 16% of patients reported pain and disability after one year [11]. Moreover, DRFs increase the risk of subsequent fractures in patients with osteoporosis [11,12].

In the current literature, a myriad of classification systems for DRFs are available [13]. In 1814, Abraham Colles first described an extra-articular pattern of DRF in which the broken end of the radius is displaced backwards [14], frequently as a result of a fall on an outstretched hand [13,14]. As regards contemporary classification systems, the AO Classification (AO—Comprehensive Classification of Fractures) is by far the most detailed classification [13,15]. However, its reproducibility and inter-observer reliability have been shown to be problematic [15,16,17].

Arnold and colleagues reported that females over 50 years of age had an increased risk of DRFs compared to the healthy population [18]. In addition, the association between DRFs and osteoporosis has been well-described for many years [5,19]. These data were confirmed by several authors who proposed a universal screening campaign with Bone Mass Density (BMD) assessment using dual-energy x-ray absorptiometry (DXA) in female patients aged over 50 years to diagnose and treat osteoporosis early [20,21,22].

The Western population’s increased ageing and life expectancy have resulted in a rising number of osteoporotic fractures, with a consequent significant burden for the healthcare system. To our knowledge, there is no international consensus regarding the screening and management of DRFs. Moreover, only a few countries have a national register reporting the fracture trends [3,23,24,25,26]. Understanding the national trends and patient characteristics of this type of fracture could make it easier to compare data between countries [27,28]. Reporting national data and statistics could allow clinicians to compare outcomes internationally [29].

In Italy, there is no universal screening campaign for osteoporosis. Reporting the results of a nation with a lack of screening campaign could be helpful when comparing the incidence of hospitalizations for DRFs (and more specifically Colles’ fractures) worldwide. The objective of this study was to report the annual incidence of hospital admissions for Colles’ fractures in Italy from 2001 to 2016, based on public and private hospital reports. The purpose of this study was to compare national data with other countries.

## 2. Materials and Methods

This paper is a nationwide registry study of all surgical procedures for Colles’ fractures diagnosed in each Italian hospital (private and public) between 2001 and 2016. Patients’ records were obtained from the National Hospital Discharge records (SDO), an official diagnoses and procedures database maintained at the Italian Ministry of Health. The present study included all patients with 813.41 ICD-9-CM code (International Classification of Diseases, Ninth Revision—Clinical Modification code); i.e., all patients with a Colles’ fracture diagnosis. However, the 813.41 ICD-9-CM is also used to code greenstick and torus fractures in young patients. To focus the attention of the present study on complete fractures and avoid overestimation of the population who may suffer from Colles’ fractures, only patients over 15 years of age were included. Each record contains the patient’s age, sex, domicile, length of the hospital stay (days), primary diagnoses and primary procedures. In addition, adult population data from the National Institute for Statistics (ISTAT) for each year were used to calculate the incidence rates.

Descriptive statistical analyses were used, frequencies and percentages for categorical data, and mean and standard deviations for continuous data. Incidence rates of Colles’ fractures were calculated using the annual adult population size obtained from ISTAT (Italian National Institute of Statistics). All statistical analyses were performed using the Statistical Package for Social Sciences (SPSS, version 26, IBM Corp., Armonk, NY, USA) and Excel (version 365, Microsoft, Redmond, WA, USA) software.

## 3. Results

### 3.1. Demographics

From 2001 to 2016, 120,932 procedures for Colles’ fractures were performed in Italy. The cumulate period of incidence was 14.8 procedures for Colles’ fractures for every 100,000 adult Italian inhabitants, observing the highest incidence between 2004 and 2006 (Figure 1).

The majority of surgeries were found in the 65–69- and 70–74-year age groups (Figure 2).

Overall, most patients were female (71.1% of women and 28.9% of men). However, patients between 15 and 49 years old were primarily male (Figure 3).

From 2001 to 2016, the average age of patients was 60.2 ± 17.6 years (65.6 ± 14 years for females and 46.8 ± 18.3 years for males, respectively).

### 3.2. Length of Hospitalization

The average length of hospitalization was 3.3 days ± 3.6 (range 0–336 days). Between 2001 and 2016, the trend of the mean number of days of hospitalization increased, with a peak in 2011 (Figure 4).

Males experienced more days of hospitalization than females (females 3.2 ± 3.5 days and males 3.6 ± 3.8 days). However, female patients aged 15–19, 95–99 and 100+ showed a higher number of mean hospital-stay days (Figure 5).

### 3.3. Main Primary Procedures

During the 16-year study period, the main primary procedures were Closed Reduction of Fracture Without Internal Fixation, Radius and Ulna (34.5%; procedure code: 79.02), Closed Reduction of Fracture With Internal Fixation, Radius and Ulna (27% procedure code: 79.12), Open Reduction of Fracture With Internal Fixation, Radius and Ulna (21.9% procedure code: 79.32), Application of External Fixation Device, Carpals and Metacarpals (9.6% procedure code: 78.14) and Application of External Fixation Device, Radius and Ulna (4.4% procedure code: 78.13) (Figure 6).

### 3.4. Economic Impact

To date, the average Italian hospital reimbursement for a “Closed Reduction of Fracture Without Internal Fixation, Radius and Ulna” ranges from EUR 191 (one-day-stay procedure) to EUR 753 (more-than-one-day-stay, with an increment of EUR 7 for every extra day of hospitalization) for each hospital admission. Overall, between 2001 and 2016, a total cost of EUR 23,154,905 has been calculated for a “Closed Reduction of Fracture Without Internal Fixation, Radius and Ulna” relating to Colles’ fracture diagnoses. The annual average cost was about EUR 1,447,181 ± EUR 869,238 with a range from EUR 301,150 in 2016 to EUR 2,739,299 in 2002 (Figure 7).

The average Italian hospital reimbursement for a “Closed Reduction of Fracture With Internal Fixation, Radius and Ulna” ranges from EUR 1590 (one-day-stay procedure) to EUR 4391 (more-than-one-day-stay, with an increment of EUR 13 for every extra day of hospitalization) for each hospital admission. Overall, between 2001 and 2016, a total cost of EUR 116,506,606 has been calculated for a “Closed Reduction of Fracture With Internal Fixation, Radius and Ulna” relating to Colles’ fracture diagnoses. The annual average cost was about EUR 7,281,662 ± EUR 1,225,726 with a range from EUR 5,038,959 in 2016 to EUR 8,907,944 in 2010 (Figure 7).

The average Italian hospital reimbursement for an “Open Reduction of Fracture With Internal Fixation, Radius and Ulna” ranges from EUR 1590 (one-day-stay procedure) to EUR 4391 (more-than-one-day-stay, with an increment of EUR 13 for every extra day of hospitalization) for each hospital admission. Overall, between 2001 and 2016, a total cost of EUR 100,823,246 has been calculated for an “Open Reduction of Fracture With Internal Fixation, Radius and Ulna” relating to Colles’ fracture diagnoses. The annual average cost was about EUR 6,301,452 ± EUR 3,374,693 with a range from EUR 1,062,270 in 2001 to EUR 10,590,468 in 2016 (Figure 7).

## 4. Discussion

A validated national registry was used to identify Colles’ fracture hospitalizations in private and public hospitals in Italy for the 15 years of the present study. Throughout the 15 years, there was a peak in the incidence of Colles’ fracture between 2004 and 2006. However, since 2006, the incidence of Colles’ fractures per 100,000 residents in Italy has been declining. These data differ from results reported in other countries where DRF incidence has tended to increase [11]. To be specific, according to Stirling and colleagues [11], an increase in DRF incidence is reported in Northern Europe, Australia, North America, Asia and Africa [11]. The differences in national distribution could be explained due to cultural or lifestyle factors, including rural living, obesity and osteoporosis, according to Porrino et al. [1].

The peak number of Colles’ fractures in Italy occurred between the ages of 60 and 79. During adolescence and young adulthood, men had a greater incidence of Colles’ fractures.

Several factors account for this distribution of incidence of Colles’ fractures, including lifestyle, comorbidities, age and sex [27]. The increasing age of the worldwide population and related comorbidities of older patients may predispose patients to bone fragility and traumas [30]. To be specific, osteoporosis and low mineral density could reduce bone density resulting in a higher fracture risk [31,32,33]. The most common causes of these fractures in young adults are playing sports, motor vehicle accidents and high-energy trauma [11]. Otherwise, the most common cause in older adults is low-energy trauma due to falling on an extended hand [11]. Post-menopausal women, osteopenia and osteoporosis (assessed by DXA) reported a higher fracture risk [11,20,27]. Stirling and colleagues reported a predicted increase in post-menopausal osteoporotic fractures by 17% [11]. Moore et al. reported 10,259 hospitalizations for DRFs in the UK population between 2007 and 2016 [27]. Moreover, the peak of incidence was recorded between 2009 and 2010 in female populations over 50 years old, confirming the present study’s data [27].

Ostergaard and colleagues proposed high-risk screening patients [34]. A screening campaign by the BMD assessment would provide a considerable benefit in reducing the cost of these fractures. However, the cost of DXA for the whole population would be significant. Future studies should assess the cost effectiveness of the DXA universal screening campaign to reduce the incidence of distal radius and other fractures. A Canadian study in 2006 reported the results of a screening campaign using DXA and the associated trends in wrist and hip fractures [35]. The authors reported that with the increased use of DXA, there was a concomitant increase in the use of antiresorptive drugs in the elderly population (over 65 years). The authors also reported a consequent decrease in hip fracture rate in this population [35]. A cheaper method for detecting osteoporosis early could be screening for vitamin D deficiency in the adult population. No organization recommends population-based screening for vitamin D deficiency [36]. Vitamin D (and calcium) is an inexpensive and easily obtainable supplement in the diet. Accordingly, recent studies have shown that primary and secondary osteoporosis have also become significant pediatric diseases [37,38,39]. Despite these considerations, a recent study stated that the available literature is insufficient to determine the advantages and disadvantages of screening asymptomatic persons for vitamin D insufficiency [36]. Other high-quality studies are required to confirm the results worldwide.

The length of hospitalization reached a peak of 3.3 days in 2011 and then progressively decreased. Usually, men reported a longer average hospital stay compared to women. Conservative treatment was the most common procedure performed.

DRFs are associated with poor quality of life, high healthcare costs and interruption in daily activities [11]. Most patients fully recover within 3–6 months [11], but 16% of patients report pain, disability and hand stiffness over the first year, resulting in a prolonged back-to-work time [10,11,40]. According to Stirling and colleagues, Moore et al. reported that 11% of UK patients with DRFs experienced moderate to severe pain one year later [27]. Similarly, it has been reported that in Canadian populations, 11% of patients also suffer from moderate to severe pain one-year post DRF [41]. Furthermore, these fractures increase the risk of future fractures in osteoporotic patients [11].

### 4.1. Economic Analysis

Chung and colleagues, in 2007, reported an average healthcare cost of USD 170 million due to DRFs [42]. In the U.S., the cost is currently approximately USD 2 billion per year [34].

Non-surgical and surgical treatments could be used depending on the fracture type; however, the latter solution is the most commonly performed [43]. To be specific, open reduction with internal fixation is the most used procedure worldwide [43]. Although this procedure is related to good outcomes and fast recovery, it is also the most expensive. Farner et al. reported a doubled increase in the incidence of internal fixation for DRFs during a 13-year study period [44]. Cummings et al. confirmed the increased trend in DRF hospitalizations with a consequent burden for the healthcare system [45]. The burden of Colles’ fractures is also relevant in Italy, as reported in the present study’s results.

### 4.2. Limitations

Administrative data from public and private hospitals were used in this research. For all procedures reported, the International Classification of Diseases, Ninth Revision (ICD-9) was adopted. Otherwise, different codes for the same surgical operation might be used with ICD-9. This coding heterogeneity could lead to an underestimation of our results. In addition, due to the ICD-9 limitation, it was impossible to discern between the different patterns of DRF.

Moreover, a limit of the present study may be the lack of outcome scores. Furthermore, patients did not receive a unique ID number in the Italian healthcare system, as hospitalizations are anonymized. This means that patients who underwent more than one surgical procedure were potentially counted twice or more.

Additionally, the ICD-9 coding was performed by surgeons; as a result, there may be individual inter-observer variations.

Lastly, comparing the findings with other countries was difficult due to the differences in healthcare systems.

## 5. Conclusions

The results of this study and the literature reported highlighted that the burden of Colles’ fractures is relevant in Italy and worldwide. However, no international consensus was reported regarding the type of treatment and the effectiveness of screening campaigns. Therefore, epidemiological studies may help provide the necessary data for establishing international guidelines concerning the indication, screening and treatment measures for Colles’ fractures.

## Figures and Tables

**Figure 1 ijerph-20-03956-f001:**
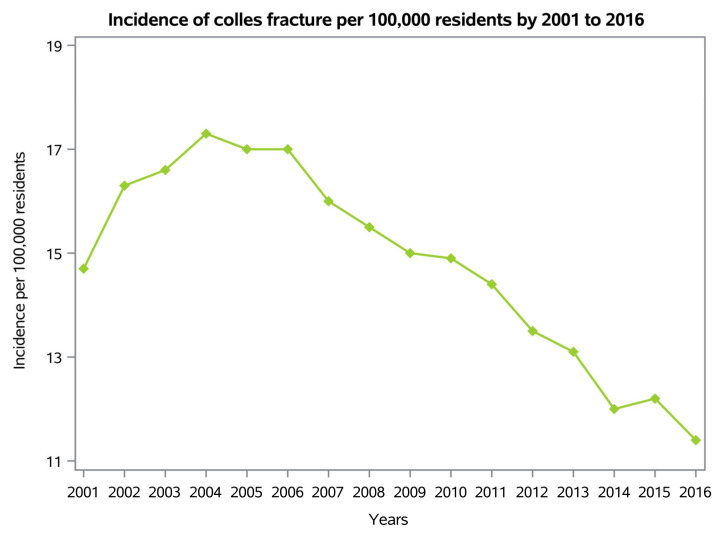
Incidence of Colles’ fractures per 100,000 residents from 2001 to 2016.

**Figure 2 ijerph-20-03956-f002:**
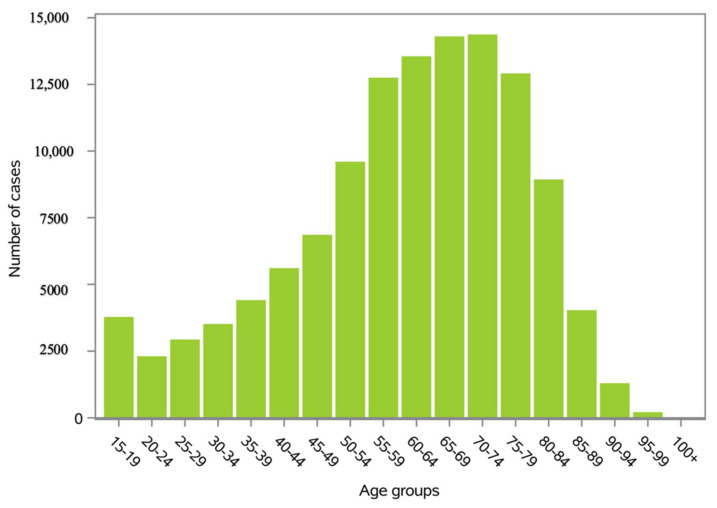
Number of Colles’ fractures by age group.

**Figure 3 ijerph-20-03956-f003:**
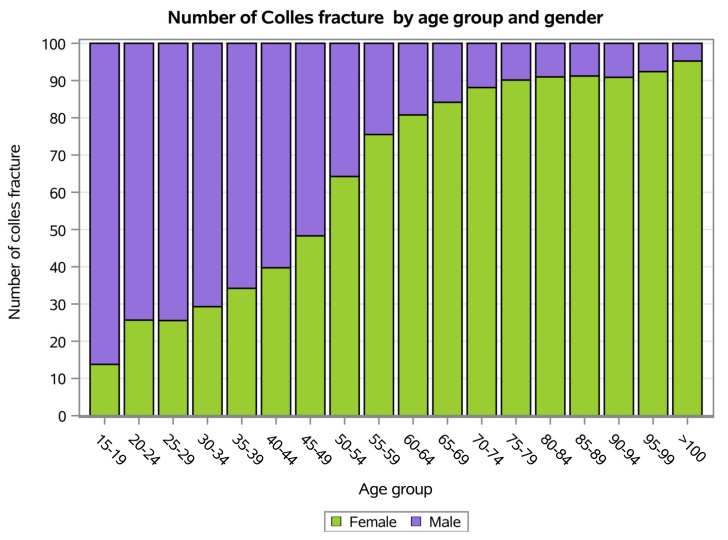
Number of Colles’ fractures by age group and gender.

**Figure 4 ijerph-20-03956-f004:**
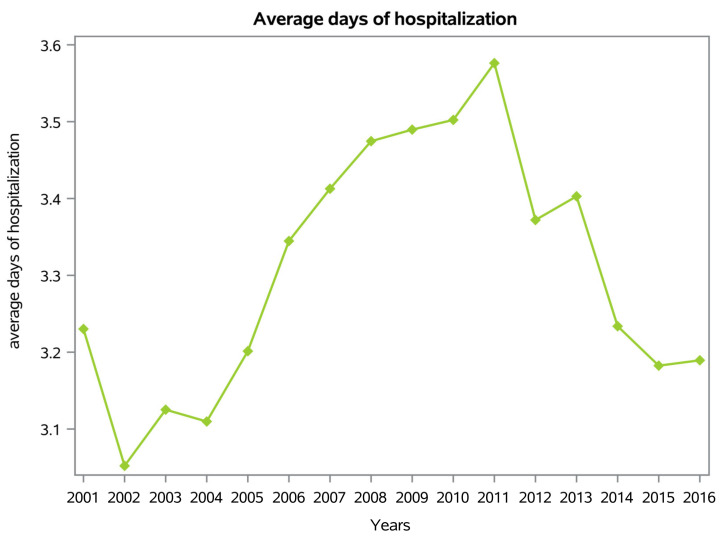
Average days of hospitalization.

**Figure 5 ijerph-20-03956-f005:**
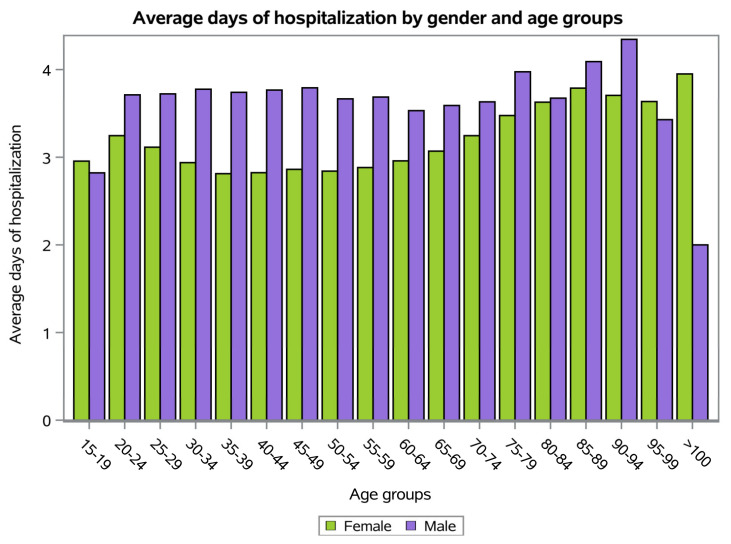
Average days of hospitalization by gender and age group.

**Figure 6 ijerph-20-03956-f006:**
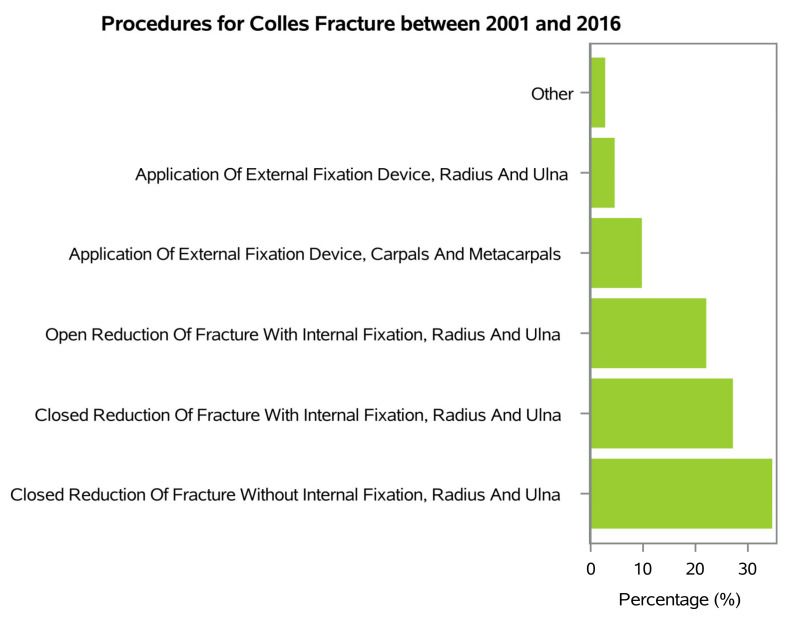
Procedures for Colles’ fractures between 2001 and 2016.

**Figure 7 ijerph-20-03956-f007:**
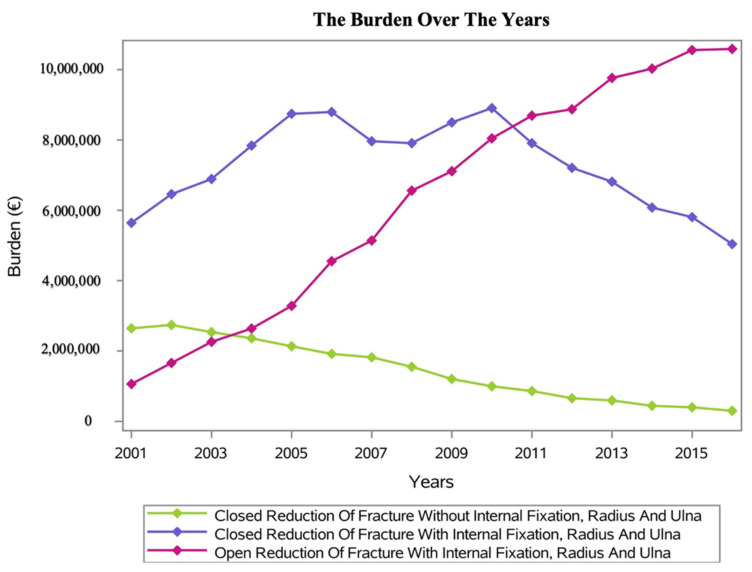
The burden over the years.

## Data Availability

The data presented in this study are available on request from the corresponding author.

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
