# Peer review of "Colles’ Fracture: An Epidemiological Nationwide Study in Italy from 2001 to 2016"

_ijerph, 2023, doi:10.3390/ijerph20053956_

Round 1

Reviewer 1 Report

Page 2 line 75 By limiting your population to those over 15 years, do you mean to avoid OVERestimation of the population with DRF?

Page 5 line 111 In parentheses, do you mean to say males 3.6 +/- 3.8 days?

You mention the possibility of instituting a nationwide BMD assessment via DXA as having potential benefit to reduction of the accumulative cost of fractures, yet acknowledge the cost is prohibitive. Here in the US, the cost of a DXA can range from $100 - $400, or more.  My colleagues and I are focused more at the level of Vitamin D (and calcium) in the diet, as vitamin D is an inexpensive and easily obtainable supplement. Indeed, in many otherwise "healthy" adolescents and young adults, some of whom may well have been included in your study, we are finding very low 25-hydroxy vitamin D levels (many have very low to nonexistent dairy intake). As a preventative measure, perhaps screening could begin earlier, and set the stage for bone health in later life?

Author Response

Dear reviewer

We would like to thank the reviewers for their helpful comments and their suggestions for improving our manuscript.  We have carefully revised our manuscript accordingly and hope that the changes are acceptable for publication. The changes made are itemized below with our comments (dark blue text) to the reviewer’s suggestions. Changes made in the text were done using track changes function of word and are highlighted in yellow in the original manuscript as well as reported below.

­

Reviewer #1

Page 2 line 75 By limiting your population to those over 15 years, do you mean to avoid OVERestimation of the population with DRF?

Thank you very much for your comment. As you suggested, there was an unintentional error that we have promptly corrected.

Page 5 line 111 In parentheses, do you mean to say males 3.6 +/- 3.8 days?

Thank you very much for your comment. As you suggested, there was an unintentional error that we have promptly corrected.

You mention the possibility of instituting a nationwide BMD assessment via DXA as having potential benefit to reduction of the accumulative cost of fractures, yet acknowledge the cost is prohibitive. Here in the US, the cost of a DXA can range from $100 - $400, or more.  My colleagues and I are focused more at the level of Vitamin D (and calcium) in the diet, as vitamin D is an inexpensive and easily obtainable supplement. Indeed, in many otherwise "healthy" adolescents and young adults, some of whom may well have been included in your study, we are finding very low 25-hydroxy vitamin D levels (many have very low to nonexistent dairy intake). As a preventative measure, perhaps screening could begin earlier, and set the stage for bone health in later life?

Thank you very much for your comment and your precious suggestions. We have now added a paragraph in our discussion section relative to the importance of Vitamin D in bone health and the possibility of an early screening for Vitamin D deficiency in the adult population, for preventing osteoporosis and consequently frailty fractures.

Reviewer 2 Report

very well-written paper

line 102 in brackets - I guess the numbers relate to males and females?

Author Response

Dear reviewer

We would like to thank the reviewers for their helpful comments and their suggestions for improving our manuscript.  We have carefully revised our manuscript accordingly and hope that the changes are acceptable for publication. The changes made are itemized below with our comments (dark blue text) to the reviewer’s suggestions. Changes made in the text were done using track changes function of word and are highlighted in yellow in the original manuscript as well as reported below.

Reviewer #2

line 102 in brackets - I guess the numbers relate to males and females?

Thank you very much for your appreciation of our manuscript. As requested, we have now better specified in the text that the average age of patients was 60.2 ± 17.6 years (65.6 ± 14 years for females and 46.8 ± 18.3 years for males, respectively).

Reviewer 3 Report

The authors examined incidence, age and sex distribution, and hospitalization days for ICD-9 813.41 Colles fractures in a retrospective analysis of Italian national hospital discharge data. Colles fractures account for most but not all fractures of the distal radius. My main criticism is that it sometimes appears that DRF and Colles fractures are used as synonyms.

I suggest
- to add a paragraph to the introduction explaining the classification of DRF
- to change DRF to Colles fracture in the title
- to use the term Colles fracture when referring to the results of your analysis (ICD-9 813.41) and DRF when talking about fractures of the distal radius in general (e.g., citing other studies)

Author Response

Dear reviewer

We would like to thank the reviewers for their helpful comments and their suggestions for improving our manuscript.  We have carefully revised our manuscript accordingly and hope that the changes are acceptable for publication. The changes made are itemized below with our comments (dark blue text) to the reviewer’s suggestions. Changes made in the text were done using track changes function of word and are highlighted in yellow in the original manuscript as well as reported below.

Reviewer #3

The authors examined incidence, age and sex distribution, and hospitalization days for ICD-9 813.41 Colles fractures in a retrospective analysis of Italian national hospital discharge data. Colles fractures account for most but not all fractures of the distal radius. My main criticism is that it sometimes appears that DRF and Colles fractures are used as synonyms.

I suggest
- to add a paragraph to the introduction explaining the classification of DRF
- to change DRF to Colles fracture in the title
- to use the term Colles fracture when referring to the results of your analysis (ICD-9 813.41) and DRF when talking about fractures of the distal radius in general (e.g., citing other studies)

Thank you very much for your comments and for the opportunity to better explain these points. Now we have corrected and improved our article with your precious suggestions.